# Attitude towards People with Disability of Nursing and Physiotherapy Students

**DOI:** 10.3390/children7100191

**Published:** 2020-10-20

**Authors:** Petronila Oliva Ruiz, Gloria Gonzalez-Medina, Alejandro Salazar Couso, María Jiménez Palomares, Juan Rodríguez Mansilla, Elisa María Garrido Ardila, María Nieves Merchan Vicente

**Affiliations:** 1Department of Nursing and Physical Therapy, University of Cadiz, 11009 Cádiz, Spain; petronila.oliva@uca.es; 2University Institute of Research in Social Sustainable Development (INDESS), University of Cadiz, 11009 Cádiz, Spain; 3Department of Statistics and Operational Research, University of Cadiz, 11009 Cádiz, Spain; alejandro.salazar@uca.es; 4Biomedical Research and Innovation Institute of Cadiz (INiBICA), Research Unit, Puerta del Mar University Hospital, University of Cadiz, 11009 Cádiz, Spain; 5Observatory of Pain, Grünenthal Foundation-University of Cadiz, 11009 Cádiz, Spain; 6ADOLOR Research Group, 06006 Badajoz, Spain; mariajp@unex.es (M.J.P.); jrodman@unex.es (J.R.M.); egarridoa@unex.es (E.M.G.A.); 7Medical and Surgical Therapy Department, Physiotherapy Field, University of Extremadura, 06006 Badajoz, Spain; mmerchanvic@gmail.com

**Keywords:** disability, nursing, physical therapy, attitude, student

## Abstract

Background: Attitudes are a component of our behaviour. Health professionals should have a global perspective of disability. They must provide treatment to people with disability and care for them, but they also should accept them with no judgements or discrimination. The general objective of this study was to know the attitude towards people with disability of nursing and physiotherapy students at the University of Cadiz. Methods: This was a descriptive, correlational, transversal and synchronous study. A total of 200 students participated in the study (91 from the bachelor’s degree in nursing and 109 from the bachelor’s degree in physiotherapy). The ‘Attitudes towards people with disability scale’ was used. Results: The mean score for both groups of students was 157.05 (SD = 14.14). Conclusions: Attitudes towards disability of nursing and physiotherapy students at the University of Cadiz tend to be positive. However, this was considered not sufficient since they will be health professionals in the future.

## 1. Introduction

The most common definition of attitude is ‘the person’s tendency or willingness to assess, in some way, an object or its symbol’ [1] and ‘to behave accordingly’ [2]. According to Braza [3], attitude is ‘the position from where a person observes a social phenomenon based on what he or she thinks and feels about it, predisposing the person to react in a specific way’.

Attitudes are a component of our behaviour. The relationship between our attitudes and our behaviour is influenced by cognitive and emotional factors as well as behaviour intention. Interpersonal behaviour is also influenced by our attitudes. Therefore, the study of attitudes is essential for health practice [4], its analysis being helpful to assess the quality of the professional performance [5].

The World Health Organisation (WHO) defines disability as a general term that includes impairments, activity limitations and restrictions on participation. In particular, impairments are related to the anatomy structure or body function [6]. 

Although the concept of disability develops socially [7], its meaning changes depending on the culture and the time. Health professionals must have a global perspective of people with disability, caring and providing treatment as well as accepting their disability and dependency situation [8]. This means that they should have no judgement or discrimination towards these persons and they must treat them, forgetting the physical independence as the main goal.

In 2015, people with disabilities accounted for 5.9% (1.774.800) of the total Spanish population [9]. Disability prevalence is expected to increase due to aging population and the United Nations (UN) recommends the implementation of national and international measures. The UN recommends an education based on tolerance and solidarity for professionals that work with people with disabilities [10]. 

Nurses and physiotherapists are essential health professionals in the multidisciplinary team as they have direct contact with the patients [11]. Moreover, the personal experience of the students of these disciplines will condition their attitude towards the patients [12,13,14]. 

Based on all this, the general objective of this study was to analyse the attitude towards persons with disability that the students of nursing and physiotherapy bachelor’s degrees of the University of Cadiz have. In addition, the specific objectives were: to compare attitudes between the students of both disciplines, to analyse the attitudes by academic year, to identify any associated factors and to verify if the type of disability influenced the attitudes of the students.

## 2. Materials and Methods

This was a descriptive, correlational, transversal and synchronous study [15].

The target population were the students of the nursing and physiotherapy bachelor´s degrees of the University of Cadiz. The selection of the sample was for convenience. All students in both grades were given the opportunity to participate. The total number of students in the academic year of 2017/2018 was 941. A total of 468 students were enrolled in the nursing bachelor’s degree in Cadiz (79.7% females vs. 20.3% males), 250 in the nursing bachelor´s degree in the locality of Jerez de la Frontera (80% females vs. 20% males) and 223 in the physiotherapy bachelor’s degree (51.1% females vs. 48.9% males). The final sample consisted of 200 subjects.

### 2.1. Assessment Tools

The ‘Attitudes towards people with disabilities scale’ (G form) was the assessment tool used [16]. It can be found in annex 1. This scale allows for the analysis of the attitudes towards disability through a score whose lower values indicate a more negative attitude and higher values indicate more positive attitudes. The ‘Attitudes towards people with disabilities scale’ has passed reliability studies (Cronbach alpha: 0.92) and validity tests (general and specific tests for physical, sensitive and mental disabilities) [8]. 

It consists of three different parts. The first part collects general data such as age and gender. The items ‘level of education’ and ‘profession’ from the original scale were substituted by ‘academic year’ and ‘Bachelor’s Degree’. The possible choices were 1st, 2nd, 3rd or 4th year and nursing or physiotherapy degrees, respectively. 

The second part assesses if the professional has any contact with persons with disability. Should that be the case, the frequency, purpose of the contact (Friendship/Leisure, Family, Work or Other) and type of disability is specified.

The third part has 37 items accompanied by a brief explanation of how to complete this part. The items that included the term ‘normal person’ were replaced by ‘people with disability’. This change was made because the term ‘normal person’ could suggest that people with disability are not normal. This part includes five factors [17]:
Validation of capacities and limitations. The items 1, 2, 4, 7, 8, 13, 16, 21, 29 and 36 refer to the subject’s view regarding the performance capacities and task implementation attitudes that a person with disability has.Personal implication. The items 3, 5, 10, 11, 25, 26 and 31 reveal the behaviour of the participant in the interaction with people with disability. A high score indicates a positive result and the acceptance of the persons with disability in personal, working and social scenarios.Rights recognition/rejection. The items 6, 9, 12, 14, 15, 17, 22, 23, 27, 35 and 37 are focused on the recognition of the human rights of people with disability and their social inclusion.Generic qualification. The items 18, 20, 24, 28 and 34 include general rating, personality features and behaviour that characterise persons with disability. The higher scores indicate a normalization of the perception of people with disability.Roles assumption. The items 19, 30, 32 and 33 refer to different ideas related to the concept that the persons with disability have about themselves.


Each item was completed according to the Likert type scale of 1 (strongly disagree) to 6 (strongly agree). The total score ranges from 37 to 185, where the lower scores indicate worse attitudes and the higher scores indicate better attitudes.

### 2.2. Procedure

All participants were informed about the fact that their involvement in the study was anonymous and voluntary. The data were collected through an online application whose link and password were provided individually to the participants. The study was reviewed and approved by the Bioethics Committee of Research of the Puerta del Mar University Hospital and Bahía de Cádiz-La Janda District (Cadiz, Andalucia, Spain) with the reference number: POR-2017. The ClinicalTrials.gov Study Identifier is NCT04498637.

Measures of frequency and central tendency for descriptive analyses were calculated, and the Kolmogorov–Smirnov test was used for testing the normality. In order to analyse the differences between academic years, chi-square tests (categorical variables) or Kruskal–Wallis tests (mean differences in the score of the attitudes scale) were used. Similarly, chi-square and Mann–Whitney tests were used to assess the differences between both degrees. A linear regression model was carried out to determine the factors related to the attitudes towards mental disability. The dependent variable was the score of the attitudes scale, and the rest of covariates were tested as potential factors. All the analyses were performed using the statistical software SPSS v21.0.

## 3. Results

Of the 200 students that constituted the sample (Table 1), 91 were enrolled in the nursing bachelor’s degree, where 21 were male (23.1%) and 70 were female (76.9%). The 62.2% of the nursing students were between 21 and 30 years old, 33% were less than 20 years old, 3.3% were between 41 and 50 years old and 1.1% were between 51 and 60 years old. Most of the students were in their first academic year (27 students–29.6%), followed by the fourth year (24 students–26.4%), the second year (21–123.1%) and the third year (19–20.9%). One hundred and nine students from the physiotherapy bachelor’s degree participated in the study. Of that number, 43 students were male (39.4%) and 66 were female (60.6%). Regarding the age, 52.3% were less than 20 years old, 42.2% were between 21 and 30, 4.6% between 41 and 50 years) and 0.9% were between 51 and 60. The highest number of students was for students in their first academic year (39 students–35.8%), followed by the third year (26 students–23.8%), the second year (23 students–21.1%) and the fourth year (21 students–20.9%).

In general, most of the students were between 18 and 30 years old. There was a greater presence of female students (63%), which coincides with the gender distribution in both degrees. The results regarding the bachelor’s degree and the academic year were mainly even. However, the majority of the participants (33%) were in their first academic year in both degrees.

It is worth highlighting that 44% of the participants had never had any contact with persons with disability. From the 55.5% who had contact with someone with a disability, 88.3% did not have contact with any person with hearing or visual impairment. A total of 71% did not have contact with any person with multiple disability, 69.4% did not have contact with mental disability and 39.6% with physical disability. The maximum score of the scale was obtained by one participant. The mean score for both groups was 157.05 (SD = 14.14).

No statistically significant differences were found between the age (*p* = 0.577) and the gender of the participants (*p* = 0.880). However, there were significant differences when comparing the gender and the frequency of the contact with persons with disability. Male participants showed the higher percentage of ‘always’ contact (16.37%), while female participants showed a higher percentage in the option ‘usually’ (38.8%). 

Regarding the differences between the bachelor’s degrees (Table 2), there was a higher number of women from the nursing degree than the physiotherapy degree. 

On the other hand, the physiotherapy students had less contact with people with hearing impairment compared with nursing students (4.8% vs. 20.4%). Between both degrees, no significant differences were found in relation to the attitude towards people with disability. When both degrees where analysed separately, there were no statistically significant differences regarding age and gender (Table 3).

The results show significant differences between the ‘academic year’ and the ‘purpose of contact’ (*p* = 0.019). The contact for family reasons was the most selected option by students of the first, third and fourth year (62.5%, 65.4% and 62.1% respectively). As can be seen in Table 4, no statistically significant differences were found in relation to the academic year, except from the frequency of contact with people with disabilities (*p* = 0.047). It is interesting to point out that the students from the initial academic years had less contact with people with disabilities than those from the more advanced academic years.

There is a significant relation between the ‘purpose of contact’ and the frequency of the contact (*p* ≤ 0.01). In the category ‘occasionally’, the option ‘friendship/leisure’ had the highest percentage (46.4%), while the option ‘other’ had the lowest with an ‘always’ frequency of contact of 0% (Table 4). 

Table 5 displays the results of the simple linear regression model. This analysis was conducted to know the attitudes towards disability related factors. The students in the three first academic years showed a worse attitude than the students from the fourth course. However, only the difference between the third and the fourth year was statistically significant (*p* = 0.043). Specifically, the students of the third year had a mean score of 7.346, which indicates a worse attitude. On the other hand, the participants that had contact with people with visual impairment had a worse attitude in comparison to those who had contact with this type of disability (8.807 points less, *p* = 0.034). The participants that had contact with persons with mental disability showed a better attitude than the participants that did not (7.382 more points, *p* = 0.007).

There is a significant relation between hearing impairment and the academic year (*p* = 0.011). The nursing students had more contact with persons with this disability, although the score in relation to the attitude obtained by the students that had contact and those who did not was similar (No 158 vs. Yes 160).

Visual impairment showed no significant relation with the attitude towards disability, although there was a difference of seven points that indicates a better attitude of the students that had contact in comparison with those that did not have it.

In general, the results suggest that having early contact with mental disability in the school years is a predisposing factor to having a negative attitude towards disability.

Multiple disability showed a significant relation with the ‘purpose of contact’ as the participants that had chosen the options ‘family’ or ‘work’ had better scores and a more positive attitude

## 4. Discussion

The attitude towards disability of the nursing and physiotherapy students of the University of Cadiz is mainly positive. 

The age and gender did not influence the attitude which coincides with the results obtained in previous studies [18,19,20,21,22].

The sample of our study showed that, in both bachelor’s degrees, female students are present in larger numbers than male students, being higher in nursing than in physiotherapy. This confirms that, in health professions such as nursing and physiotherapy, there is more female staff [23,24,25], which has been described as the ‘feminisation of health professions’ [26,27,28,29,30]. 

Regarding gender/frequency/purpose of contact with disability, male participants had an always family contact while female had more usual family contact. 

Comparing the degree and the attitude, no statistically significant differences were found between nursing and physiotherapy students. The scores were positive although they did not reach the expected result. The figures obtained by the nursing students coincide with other authors [31], who conclude that the students do not have a positive attitude towards people with disability but the attitude improves from the third year onwards. No studies conducted with physiotherapy students in Spain were found in the literature. We consider that this information is important for our society, where life expectancy is leading to a higher prevalence of people with disabilities who need health care [32].

Our results in relation to the attitude of the nursing students coincide with those from other countries. Thus, the study conducted by Horner [12] in Japan or the one conducted by Sahin and Akiol [33] in Turkey demonstrated that the students have moderate positive attitudes. In the United States, Tervo (2004) affirmed that Health Professional students have less positive attitudes than the general population and that nursing students have better attitude than the students from other health professions [34]. The study carried out by Ten et al. [35] showed that nursing students from Holland have better attitude towards people with physical and mental disability than other students. In contrast, a study from Greece [36] revealed that nursing students had a bad attitude. Regarding physiotherapy, only one study was found in the literature. This study [37] showed that physiotherapy students have a better attitude than nursing students but worse than occupational therapy students. 

In addition, our study shows that physiotherapy students had more contact with people with physical, visual, mental and multiple disability than nursing students who had more contact with people with hearing impairment. The main purpose of the contact of the nursing students was ‘work’, while the physiotherapy students had contact for ‘family’ and ‘other’ purposes.

With regard to the academic year, the students from the first years in both degrees had a more negative attitude compared to the students from the last years. Although this difference was not statistically significant, it could be due to the fact that the students in the first years have not done their clinical placements yet. This means that if they have not had any contact with people with disability in the family or friends circle, they may not know what disability is in reality. This unfamiliarity leads to disability unawareness and it has been demonstrated that unawareness increases prejudices, fear [38,39] and therefore negative attitudes [25,40,41]. The students from the last years, who had completed their clinical placements, showed a more positive attitude, which was statistically significant. These results coincide with other studies [42,43]. Moreover, some authors have also revealed that previous contact or experiences with people with disability can lead to better attitudes towards them [33,34,35] and consequently would decrease the presence of stereotypes [44]. Our results suggest that the contact with people with disabilities must be encouraged in order to decrease prejudices, rejection and stigmatisation towards them. We suggest increasing welfare practices or training visits in disability centres and expanding these from the first training courses. This must be done not only at university but also in primary and secondary schools as well as throughout society.

Factors related to a worse attitude towards disability included: no contact with people with disability during the current academic year, and contact with people with mental disability.

On the other hand, our study reveals a significant relation between the purpose and the frequency of the contact. The ‘family/leisure’ option with an occasional frequency had a more positive attitude than a family contact with an ‘always’ frequency. This last option showed a more negative attitude which would be interesting to analyse in further research studies.

The related factors with a worse attitude towards disability were the academic year not having contact with people with disability and having contact with people with mental disability. Some authors affirm that having contact with a specific type of impairment influences the attitude towards people with disability [45]. In addition, contact with people with mental disability is a predisposing factor for the development of a negative attitude towards disability in general [36].

Based on all this, we can conclude that nursing and physiotherapy students have a worse attitude towards people with mental disability than towards people with other types of impairments. Those who had a previous interaction with someone with a physical disability show a better attitude towards mental disability. However, if the contact was before with a person with mental disability, they show a more negative attitude towards physical disability. Moreover, the contact with a person with a disability that involves severe communication difficulties and a stereotyped non-verbal communication such as mental disability could cause a negative attitude towards disability [46,47]. We assume that if there are communication difficulties, understanding becomes complicated. Based on the literature, we believe attitudes towards disability would improve with interaction and information [48]. Further studies would be recommended to clarify this possible relation.

Regarding hearing impairment, the physiotherapy students showed a less positive attitude towards people with this disability than nursing students. That could be due to the fact that physiotherapists do not normally treat people with hearing impairment during their clinical placements [34,45].

Multiple disability had a significant relation with the ‘purpose of contact’ being more positive between students who had contact for family and working reasons.

Based on our results, we can affirm that negative attitudes towards disability rely on the lack of interaction with people with disability or on the lack of information [49]. In other words, disability awareness creates a more appropriate attitude and decreases rejection and stigmatisation.

## 5. Conclusions

Attitudes towards disability of nursing and physiotherapy students at the University of Cadiz are generally positive. However, this was considered as not sufficient since they will be health professionals in the future who will treat and care for people with disabilities and their inclusion. Therefore, they should have obtained the maximum score.

Attitudes are more negative among the students who did not have any contact with people with disability (in the first academic years) or had contact with people with mental disability as their first disability experience.

## Figures and Tables

**Table 1 children-07-00191-t001:** Characteristics of the participants.

Variable	Category	N	%
Age	<20 years	87	43.5
	21–30 years	103	51.5
	31–40 years	0	0
	41–50 years	8	4
	51–60 years	2	1
Gender	Male	66	33
	Female	134	67
Bachelor’s Degree	Nursing	91	45.5
	Physiotherapy	109	54.5
Academic year	First	66	33
	Second	44	22
	Third	45	22.5
	Fourth	45	22.5
Have you had contact with people with disability?	Yes	111	55.5
	No	89	44.5
Purpose of the contact (N = 110)	Friendship/Leisure	28	25.5
	Family	66	60
	Work	11	10
	Other	5	4.5
Frequency of the contact (N = 110)	Occasionally	24	21.8
	Frequently	23	20.9
	Usually	32	29.1
	Always	31	28.2
Physical Disability (N = 111)	Yes	67	60.4
	No	44	39.6
Hearing Impairment /Disability (N = 111)	Yes	13	11.7
	No	98	88.3
Visual Impairment/Disability (N = 111)	Yes	13	11.7
	No	98	88.3
Mental Disability (N = 111)	Yes	34	30.6
	No	77	69.4
Multiple Disability (N = 111)	Yes	32	28.8
	No	79	71.2
Range of scores of attitudes towards disability	Minimum	115	
	Mean (SD)	157.05 (14.14)	
	Maximum	185	

SD: Standard Deviation; N = sample.

**Table 2 children-07-00191-t002:** Comparison between groups.

		Bachelor´s Degree		*p*
		Nursing N = 91	Physiotherapy N = 109	
Variable	Category	*n* (%)	*n* (%)	
Gender	Male Female	21 (23.1) 70 (76.9)	45 (41.3) 64 (58.7)	0.006 a
Academic year	First Second Third Fourth	27 (29.7) 21 (23.1) 19 (20.9) 24 (26.4)	39 (35.8) 23 (21.1) 26 (23.9) 21 (19.3)	0.581 a
Have you had contact with people with disability?	Yes No	49 (53.8) 42 (46.2)	62 (56.9) 47 (43.1)	0.667 a
Frequency of the contact	Occasionally Frequently Usually Always	6 (12.5) 11 (22.9) 15 (31.3) 16 (33.3)	18 (29.0) 12 (19.4) 17 (27.4) 15 (24.2)	0.213 a
Physical Disability	Yes No	31 (63.3) 18 (36.7)	36 (58.1) 26 (41.9)	0.578 a
Hearing Impairment /Disability	Yes No	10 (20.4) 39 (79.6)	3 (4.8) 59 (95.2)	0.011 a
Visual Impairment/Disability	Yes No	6 (12.2) 43 (87.8)	7 (11.3) 55 (88.7)	0.877 a
Mental Disability	Yes No	13 (26.5) 36 (73.5)	21 (33.9) 41 (66.1)	0.405 a
Multiple Disability	Yes No	15 (30.6) 34 (69.4)	17 (27.4) 45 (72.6)	0.712 a
Attitudes towards disability Scale	Significance (SD)	157.4 (13.9)	156.7 (14.4)	0.890 b

a: Chi-square; b: Mann–Whitney’s U; N = sample

**Table 3 children-07-00191-t003:** Difference of attitudes according to gender and age in the nursing and physiotherapy bachelor´s degrees.

	Variable	Category	Attitude Score		
			Mean	SD	*p*
Nursing	Gender	Male	160.62	14.17	0.179 b
		Female	156.46	13.8	
	Age	<20	156.7	10.23	0.349 c
		21–30	158.4	14.83	
		31–40	*	*	
		41–50	155.33	24.44	
		51–60	129	**	
Physiotherapy	Gender	Male	155.4	15.18	0.475 b
		Female	157.69	13.84	
	Age	<20	155.11	14.16	0.225 c
		21–30	157.87	14.98	
		31–40	*	*	
		41–50	160.4	7.16	
		51–60	180	**	

SD: Standard Deviation. b Mann–Whitney´s U. c Kruskal–Wallis’ H. * Data not available for this category. ** Unable to calculate SD as only one record is available.

**Table 4 children-07-00191-t004:** Comparisons by academic year.

		Academic Year				*p*
		1st Year N = 66	2nd Year N = 44	3rd Year N = 45	4th Year N = 45	
Variable	Category	*n* (%)	*n* (%)	*n* (%)	*n* (%)	
Gender	Male Female	21 (31.8) 45 (68.2)	16 (36.4) 28 (63.6)	17 (37.8) 28 (62.2)	12 (26.7) 33 (73.3)	0.671 a
Bachelor´s Degree	Nursing Physiotherapy	27 (40.9) 39 (59.1)	21 (47.7) 23 (52.3)	19 (42.2) 26 (57.8)	24 (53.3) 21 (46.7)	0.581 a
Have you had contact with people with disability?	Yes No	32 (48.5) 34 (51.5)	24 (54.5) 20 (45.5)	26 (57.8) 19 (42.2)	29 (64.4) 16 (35.6)	0.410 a
Purpose of the contact	Friendship/ Leisure Family Work Other	8 (25.0) 20 (62.5) 1 (3.1) 3 (9.4)	11 (47.8) 11 (47.8) 0 (0.0) 1 (4.3)	4 (15.4) 17 (65.4) 5 (19.2) 0 (0.0)	5 (17.2) 18 (62.1) 5 (17.2) 1 (3.4)	0.019 d
Frequency of the contact	Occasionally Frequently Usually Always	N = 31 10 (32.3) 8 (25.8) 8 (25.8) 5 (16.1)	N = 24 9 (37.5) 3 (12.5) 7 (29.2) 5 (20.8)	N = 26 2 (7.7) 5 (19.2) 11 (42.3) 8 (30.8)	N = 29 3 (10.3) 7 (24.1) 6 (20.7) 13 (44.8)	0.047 a
Physical Disability	Yes No	N = 32 17 (53.1) 15 (46.9)	N = 24 18 (75) 6 (25)	N = 26 18 (69.2) 8 (30.8)	N = 29 14 (48.3) 15 (51.7)	0.140 a
Hearing Impairment/Disability	Yes No	N = 32 1 (3.1) 31 (96.9)	N = 24 3 (12.5) 21 (87.5)	N = 26 6 (23.1) 20 (76.9)	N = 29 3 (10.3) 26 (89.7)	0.121 d
Visual Impairment/Disability	Yes No	N = 32 4 (12.5) 28 (87.5)	N = 24 2 (8.3) 22 (91.7)	N = 26 4 (15.4) 22 (84.6)	N = 29 3 (10.3) 26 (89.7)	0.879 d
Mental Disability	Yes No	N = 32 12 (37.5) 20 (62.5)	N = 24 10 (41.7) 14 (58.3)	N = 26 7 (26.9) 19 (73.1)	N = 29 5 (17.2) 24 (82.8)	0.195 a
Multiple Disability	Yes No	N = 32 7 (21.9) 25 (78.1)	N = 24 7 (29.2) 17 (70.8)	N = 26 6 (23.1) 20 (76.9)	N = 29 12 (41.4) 17 (58.6)	0.334 a
Attitudes towards disability Scale	Mean (SD)	156.3 (13)	157.1 (14.7)	156.2 (13.8)	159 (15.7)	0.741 c

a Chi-square. d Likelihood ratio. c Kruskal–Wallis’ H.

**Table 5 children-07-00191-t005:** Factors related to attitudes towards disability.

	B1	IC 95%	*p*
Academic Year First Second Third Fourth *	−6.308 −3.399 −7.346	(−13.045; 0.429) (−11.354; 4.556) (−14.453; −0.239)	0 0.066 0.402 0.043
Visual Impairment/Disability No Yes *	−8.807	(−16.968; −0.645)	0.034
Mental Disability No Yes *	7.382	(2.002; 12.761)	0.007

Dependent variable: Attitudes towards disability Scale * Category of reference 1 B: Beta.

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
