# Peer review of "Attitude towards People with Disability of Nursing and Physiotherapy Students"

_children, 2020, doi:10.3390/children7100191_

Round 1

Reviewer 1 Report

This is a very interesting work that could be of interest for many readers.

The goal of this review is clearly described.

The aim are interesting. Material and methods are well explained.

The introduction of the study clearly sum up the aim of the study. The authors provide a full view for performing the study based on a review of the medical literature.

The results are reported clearly and concisely. In discussion authors compare the results with other data reported in the literature. The references are qualified and follow the format for the journal.

The tables sum up the study are clearly. 

For the statistical analysis is not needed further checking of data by a statistician reviewer.

The authors provide a full view for performing the study based on a review of the medical literature.

Conclusions were almost correct and will prove useful to other researchers in the field.

The authors added caution about their study limitations. They pointed out three – one interpretation and two methodological. What successfully escalated the value of this paper according to the Evidence Based Medicine guidelines. The topic of this manuscript falls within the scope of Children

Reviewer 2 Report

This is a very interesting work that could be of interest for many readers. The results provide an advance in current knowledge (especially in Spain and in Nursing and Physiotherapy's education system). 

I think that, before publication, this work needs a minor revision.

Article is written in an appropriate way and respects the rules of the journal. The background and the aim are interesting (the introduction of the study clearly sum up the aim of the study). The authors provide a view for performing the study based on the literature (especially spanish lit.) Please add a refference after sentences in line 48-52.

The question is original and well defined. 

In the methods is present the type of study. Setting and population are well explained, but there is no information how participants were enrolled to final sample- randomly? Please explain in line 73. 

For the statistical analysis is not needed further checking of data by a statistician reviewer.

The data and analyses are presented correctly and robust enough to draw the conclusions. The highest standards for presentation of the results are used (the results are reported clearly and concisely). The results are interpreted appropriately and they are significant. Conclusions are justified and supported by the results.

The study is correctly designed and analysed with the highest technical standard. In my opinion, another researcher are allow to reproduce the results. Discussion compare the results obtained point by point with other data reported in the literature and those reported are discussed in depth explaining clearly the implications of the results.

In my opinion, the conclusion are interesting for the readership of the Journal- attitude to disabilities are very important in health care proffesionals, and more researching in this are could be done in the future. There is an overall benefit to publishing this work- the authors have addressed an important long standing conclusion. It is interesting how Academics Teachers could work with students to make their attitudes towards people with disability- this is the long standing question for all (Authors may discuss better this point in line 244-246).

The references are qualified. The reference list follow the format for the journal.
